# Numerical Simulation of Mold Filling of Polymeric Materials with Friction Effect during Hot Embossing Process at Micro Scale

**DOI:** 10.3390/polym16101417

**Published:** 2024-05-16

**Authors:** Faleh Rabhi, Gang Cheng, Thierry Barriere

**Affiliations:** 1Institut FEMTO-ST, National Centre for Scientific Research (CNRS), University of Franche-Comté, 25000 Besançon, France; 2INSA Centre Val de Loire, Laboratory of Mechanics Gabriel Lamé (LaMé), 41034 Blois, France

**Keywords:** hot embossing process, thermoplastic polymer, mechanical behavior laws, viscoplastic model, numerical modelling, friction coefficient

## Abstract

The filling efficiency during the hot embossing process at micro scale is essential for micro-component replication. The presence of the unfilled area is often due to the inadequate behavior law applied to the embossed materials. This research consists of the identification of viscoplastic law (two-layer viscoplastic model) of polymers and the optimization of processing parameters. Mechanical tests have been performed for two polymers at 20 °C and 30 °C above their glass transition temperature. The viscoplastic parameters are characterized based on stress–strain curves from the compression tests. The influences of imposed displacement, temperature, and friction on mold filling are investigated. The processing parameters are optimized to achieving the complete filling of micro cavities. The replication of a micro-structured cavity has been effectuated using this process and the experimental observations validate the results in the simulation, which confirms the efficiency of the proposed numerical approach.

## 1. Introduction

As the demand for better microstructural control is increasing continuously, it is necessary to develop an efficient process to manufacture micro-structured component at lower cost in less time. Hot embossing (HE) consists of heating the polymer substrate to an adequate temperature and compressing it with an imposed pressure or displacement. The micro features in the mold are embossed on to the polymer surface. It is a suitable process for the accurate replication of high-quality features with high aspect ratios, high porosity, and a large surface area at the micro/nano scale [1]. Polystyrene, poly (methyl methacrylate) (PMMA) and polycarbonate (PC) are polymeric materials mainly employed in HE. The main processing parameters are pressure, forming temperature, time, and demolding temperature [2].

Kim et al. investigated acoustic wave detection using a PMMA-based device obtained via HE [3]. A polymer layer was successfully replicated with high accuracy to improve the performance of a tunneling sensor. Deshmukh et al. optimized the processing parameters, such as embossed time, pressure, and temperature to manufacture a microchannel with polymer substrate [4]. Generic algorithm and Taguchi method were applied to achieve high replication accuracy approximately 96%. An in situ observation method was proposed to capture the filling ratio of the microchannel using HE [5]. The effects of various parameters on replication efficiency were investigated to satisfy its reproducibility on the PMMA microstructures. Liu et al. studied the deformation of PMMA in the filling stage of micro HE [6]. The replication precision was significantly improved during the stress relaxation and deformation recovery stage. Wang et al. improved HE to replicate the inverted pyramid microstructure array on a PMMA surface, which demonstrated high replication accuracy and good hydrophilic effect [7].

Chang and Yu introduced an ultrasonic-assisted HE process to fabricate the micro lens arrays with a PMMA substrate [8]. The obtained microstructures were confirmed to be satisfactory after the measurement of geometrical and optical properties. This simple and rapid method has potential for the mass-production manufacturing of micro lens arrays with low cost and high accuracy. Yang et al. used the carbon dioxide gas in HE to achieve a large-area replication with a PMMA substrate [9]. The main advantages of this innovative gas-assisted embossing technique were lower processing temperature and pressure, associated with a more uniform pressure distribution. Wu et al. confirmed the efficiency of a gas-assisted HE process in the replication of a PMMA micro lens array [10]. Less residual stress was observed on the replicated substrate because of the lower embossing temperature and pressure. Çoğun et al. studied the effects of HE processing parameters, such as force, time, and temperature, on the filling rate of microfluidic channels [11]. The experimental tests were carried out using the micro-milled aluminum mold with various widths. However, only the effects of the hardness of polymer substrate as a function of temperature were considered in the numerical simulation. Other mechanical properties of the polymer substrate and the suitable material behavior laws are still lacking in the optimization of HE.

The thermoplastic polymers employed in HE are embossed above their glass transition temperature (Tg). The polymers exhibit highly temperature dependent properties, including elastic, viscous, and plastic behaviors. The complex mechanical properties should be characterized in the temperature range above Tg.

A two-layer viscoplastic model (TLVP) is available to describe viscoplastic behavior of a polymer at high temperature [12]. Kichenin et al. used it to carry out the numerical simulation of the cyclic pressure test on a polyethylene specimen [13]. Bianca et al. employed the TLVP model to predict the mechanical behavior of a thermoplastic polymer under various strain rates, specifically for the description of its viscoplastic properties [14,15]. Its parameters were identified based on the stress–strain data obtained in the tensile test. The TLVP model has been applied to predict the relaxation stress vs. time for polymers at various strain rates [16]. Tensile tests were performed to characterize the material parameters. Rabhi et al. used this model to determine the rheological parameters of PMMA and studied the effect of elastic–viscoplastic behavior on mold filling via HE [17]. Abdel et al. employed the model to describe the deformation of PMMA over a large temperature zone under Tg [18]. This model was observed to offer better consistency with the experimental data than the elastoplastic one.

The simulation of HE is performed to optimize the replication of micro-components. Yun and Kim proposed a computational analysis approach to efficiently determine the HE processing parameters [19]. A constitutive model considering the work hardening and strain softening was applied to describe the thermomechanical behavior of PMMA. The finite element method (FEM) was selected, associated with an arbitrary Lagrangian–Eulerian remeshing approach, to carry out the simulation of HE. Gomez et al. investigated the mold filling of a PMMA micro structure in HE [20]. A 2D deformable geometric substrate was created to model the polymer flow under various embossing temperatures and forces. The numerical results were validated by the experimental observations in the same processing conditions. Guo et al. effectuated the numerical simulation of the demolding stage in HE using FEM [21]. The thermal stress distribution of a replicated PMMA sample during cooling and demolding was analyzed, which indicated that the friction between the mold and PMMA sample affected the polymer deformation. Kiew et al. proposed a constitutive model to characterize the PMMA behavior during HE [22]. The effects of embossing time were studied and confirmed with the experimental results. The frictional effects during HE on the mold filling were not exploited. Wang et al. employed a generalized Maxwell model for describing the viscoelasticity of an amorphous polymer in HE [23]. The relaxation tests at various temperatures were effectuated to identify the material parameters. According to the numerical simulation and experimental investigation, enough embossing time was required for filling completely the microstructures with a large cross-sectional area. Mondal et al. studied the deformation mechanism of PMMA with a generalized Maxwell model considering the contact friction and thermal expansion effects [24]. The experimental tests were effectuated and compared with simulation results to validate the numerical approach. A parametric optimization was performed to improve the filling depth of the replicated polymer.

Cheng et al. investigated HE to develop microfluidic devices [25]. Their study included an identification of material behavior and filling simulation of the process. PMMA and PC are the most appropriate amorphous thermoplastic polymers for HE. PC is often employed in mechanical engineering due to its excellent mechanical properties [26].

This paper is a continuation of the work of Rabhi et al. [17], studying the effects of viscoplastic behavior on the mold filling accuracy of micro HE. The main objective of this research is to propose a suitable viscoplastic behavior law of polymers and improve the simulation efficiency of HE. Mechanical compression tests have been performed to characterize the properties of PC and PMMA above their Tg. The TLVP model has been identified by an inverse method to determine the behavior law of the selected polymers. The contact friction is often ignored in the numerical simulation of HE. It is important to include the interface friction between the polymer substrate and mold to improve the process simulation accuracy. The effect of the friction coefficient on the mold filling was analyzed at micro scale. A series of experimental tests have been performed to confirm the numerical simulation results.

## 2. Manufacturing Process, Experimental Tests and Behavior Law

### 2.1. Description of HE

In the case of HE, a polymer plate is pressed by a micro-structured tool above Tg. It is maintained for a certain time, and cooled to a temperature under Tg to separate from the mold [27]. Figure 1 shows the steps of HE to create micro-structured components [28]. The micro-structure from the mold is transferred on to the polymer plate by the application of optimized temperature and pressure. The polymer plate exhibits viscoelastic or viscoplastic behaviors during HE. The characterization of the mechanical behavior of the polymer is essential to identify the suitable processing parameters in HE. An appropriate behavior law is also indispensable to accurately simulate the process.

HE is applied to replicate different micro patterns in various thermoplastic polymers. Figure 2 shows different geometrical shapes at micro scale obtained with PMMA [29]. Circular patterns with a diameter lower than 0.001 mm, threadlike lines with a lateral width of 0.025 mm, and hexagonal shapes with s side length of 0.015 mm and 0.003 mm spacing between the hexagons were successfully obtained.

The main advantage of HE is the possibility to create a device with a large surface and to obtain a complex structure at micro/nano scale, such as an optical sensor, diffractive lens, microfluidic channel, and so on. Tan et al. focused on the state of art creation of optical Fresnel lenses, and micro HE exhibited high accuracy compared to other manufacturing processes [30]. Jiang et al. improved the embossing process with the optimized processing parameters to produce a microfluidic channel [31]. Fewer mold tools are required in HE, and it is relatively easy to carry out [32]. Compared with conventional processes, less residual stress is obtained in the replicated substrate because of the lower processing temperature. The principal challenge in HE is to optimize the embossing temperature and pressure to achieve the full mold filling with the selected polymer.

### 2.2. Differential Scanning Calorimetry (DSC)

The thermal properties of the embossed polymer need to be characterized in order to optimize the processing temperatures. DSC is used to identify the thermal behavior of two thermoplastic polymers (PC, PMMA) and determine their Tg. In this temperature range, the state of the polymer changes from glassy to rubbery. The embossing temperature of HE is slightly above the Tg of the replicated polymer. Tg is an important parameter used to ensure a high-quality replication without defects in HE. In this research, the DSC tests were performed from 25 to 295 °C (heating rate 2 °C/min). Two heating–cooling cycles were carried out for each analysis. The first cycle was used to eliminate the thermal history of the polymers (humidity, residual stress). The second cycle was retained for the measurement of Tg. The values of Tg of PMMA and PC are summarized in Table 1.

### 2.3. Description of TLVP Behavior Law

TLVP behavior law involves elastic, plastic, and viscous parts. It consists of an elastoplastic branch in parallel with a viscoelastic one, represented in Figure 3 [33]. The elastoplastic branch consists of linear hardening, connected in parallel with the viscoelastic branch consisting of the Maxwell model. σ is stress, ε is strain and $\sigma_{Y}$ is the initial yield stress.

σ is calculated by the sum of elastoplastic stress $\sigma_{p}$ and viscoelastic stress $\sigma_{v}$, represented as follows:(1)$$\sigma =\sigma_{p}+\sigma_{v},$$

The true stress–strain in the elastoplastic branch is described as the following equations:(2)$${\sigma_{p}=E}_{p}\times \varepsilon \;\mathrm{if}\;\sigma_{p}\leq \sigma_{Y},$$
(3)$$\sigma_{p}=H\times \varepsilon^{n_{1}}\;if\;\sigma_{p}\geq \sigma_{Y},$$
where $E_{p}$ is elastic modulus in the elastoplastic branch and $n_{1}$ is the work hardening exponent. Hardening parameter H is defined as [34]:(4)$$H\times \varepsilon^{n_{1}}=\frac{\sigma_{Y}}{\varepsilon_{Y}^{n_{1}}}(\varepsilon_{Y}+\varepsilon^{pl}{)}^{n_{1}}=\sigma_{Y}(1+\frac{\varepsilon^{pl}}{\varepsilon_{Y}}{)}^{n_{1}}=\sigma_{\gamma }(1+\frac{E_{p}}{\sigma_{Y}}\varepsilon^{pl}{)}^{n_{1}},$$
where $\varepsilon_{Y}$ is strain at $\sigma_{Y}$ and $\varepsilon^{pl}$ is the plastic strain.

Viscoelastic stress $\sigma_{v}$ is expressed as:(5)$$\sigma_{v}=A^{-\frac{1}{n_{2}}}{\dot{\varepsilon }}^{\frac{1}{n_{2}}},$$

Time-hardening m is set to zero, in the case that the strain rate $\dot{\varepsilon }$ is independent of time:(6)$$\dot{\varepsilon }=A\sigma_{v}^{n_{2}}t^{m},$$
where A represents strength coefficient, $n_{2}$ represents strain hardening of Norton–Hoff law (creep strain rate = $A\sigma^{n_{2}}$), and t is time.

The total modulus is defined as:(7)$$E=E_{p}+E_{v},$$
where $E_{v}$ is the elastic modulus in the viscoelastic branch.

f is the proportion of elasticity in the viscoelastic branch to the total elasticity, which is calculated as:(8)$$f=\frac{E_{v}}{E_{p}+E_{v}},$$

The elastic strain $\varepsilon^{el}$ consists of viscoelastic component $\varepsilon_{v}^{el}$ and elastoplastic component $\varepsilon_{p}^{el}$:(9)$$\varepsilon^{el}=f\varepsilon_{v}^{el}+(1-f)\varepsilon_{p}^{el},$$

### 2.4. Uniaxial Compression Test

Uniaxial compression tests were carried out to characterize the mechanical properties of PC and PMMA. The polymer specimens were fabricated by injection molding process. They were heated to anneal for 2 h before the compression test. The cylindrical polymer specimen was compressed between two metallic molds in an oven, as shown in Figure 4. The tests were carried out at Tg + 20 °C and Tg + 30 °C and repeated five times to be consistent with the loading conditions.

### 2.5. Description of Simulation Approach

The aim of the simulation was to investigate the effects of the mechanical behavior of the material at micro scale on mold filling by taking into account its viscoelasticity and elastoplasticity. The simulation of HE was performed with two selected materials (PC and PMMA substrates with a thickness of 1 mm) using FEM, as demonstrated in Figure 5. The elastoplastic and viscoelastic parameters were determined with the results obtained in the mechanical tests. The influence of friction parameter µ between the mold and the polymer on the filling ratio was investigated in the current work.

A 2D axisymmetric substrate was created to simulate the displacement of polymer in the micro cavity, as shown in Figure 6. The mold was cylindrical, with a radius of 0.1 mm and a height of 0.05 mm, which was considered as a rigid body due to its much higher stiffness. The polymer substrate with 1 mm height was placed under the mold. The polymer substrate was treated as a deformable solid, described by a TLVP behavior law. The Coulomb friction model was proposed to describe the contact friction between the mold and the substrate. A constant friction coefficient was applied with the penalty friction formulation. The boundary conditions were applied with a constant imposed displacement of the mold and a zero-displacement imposed on the bottom surface of polymer substrate.

The mesh applied on the polymer plate consists of free triangular and quadratic elements. The total number of elements was 3322 with a minimal size of 0.002 mm in the contact area between tool and polymer, as shown in Figure 7. The CPU time of the simulation is around 600 s with a 2.4 GHz processor and 8 GB random access memory.

## 3. Characterization Results of PMMA and PC

### 3.1. Results of Compression Tests

The uniaxial compression tests were conducted with PMMA and PC to determine the viscoplastic behavior parameters. To determine the plastic behaviors of these two studied polymers, a displacement rate of 0.54 mm s^−1^ was imposed on the samples at two temperatures (Tg + 20 °C and Tg + 30 °C). The true stress σ and strain ε were determined with the following equations:(10)$$\sigma =\frac{F}{S}=\frac{{F\;\times \;H}_{1}}{S_{0}{\;\times \;H}_{0}},$$
where $H_{0}$ is the initial height, $S_{0}$, S are the initial and final cross section, respectively, and $H_{1}$ is final height after the compression test.
(11)$$\varepsilon =ln\left(\frac{H_{0}}{H_{1}}\right),$$

Figure 8 and Figure 9 illustrate the true stress–strain curves for the PMMA and PC samples.

For PMMA, the true stress increases with strain with a maximum value of 1.49 MPa for Tg + 20 °C and 0.5 MPa for Tg + 30 °C. The true stress decreases with temperature at the same true strain. The obtained true stress–strain curves of PMMA are coherent with the results obtained by Federico et al. [35]. For PC, the true stress increases with strain, up to a maximum value of 0.9 MPa for Tg + 20 °C and 0.95 MPa for Tg + 30 °C. At the same true strain, stress decreases with temperature.

The plastic deformation of PMMA is more important than that of PC at the same temperature range. PC exhibits lower mechanical properties and viscosity. The obtained true stress–strain curves are coherent with the results obtained by Tang et al. [36]. At Tg + 20 °C, the true stress–strain curve of PMMA has two domains, an elastic domain up to 2% deformation corresponding to a stress of 0.7 MPa and a plastic domain after 2% deformation. At Tg + 30 °C, the transition from elastic state to plastic state is not significant. For PC, the true stress–strain curves at different temperatures show the same trend and consist of a small elastic range and an exponential increase in the plastic state. The elastic and plastic parameters of the polymers were identified from the experimental data and are summarized in Table 2.

### 3.2. Identification of TLVP Parameters

The inverse method was applied with an optimization procedure to minimize the sum of squares of deviations between the uniaxial compression test data and the actual one. The assessment function E(X), representing the average of least squares errors, was defined as follows [15]:(12)$$\underset{X\epsilon \Omega }{\min}\left(E(X)=\sqrt{\sum_{i=1}^{N}\frac{{\left\|y_{i}^{m}\left(X\right)-y_{i}^{c}\left(X\right)\right\|}^{2}}{N}}\right),$$
where X is an unknown vector that collects a, b, and c parameters, $y_{i}^{c}\left(X\right)$ denotes the calculated true stress–strain response, $y_{i}^{m}\left(X\right)$ is the measurement data from experimental tests for vector X, N is the parameter number, and Ω is the analysis space for X.

The relaxation tests were performed to identify the viscoelastic behavior of the polymer above its Tg. A strain rate $\dot{\varepsilon }$ = 0.03 s^−1^ was imposed on the sample up to a final strain level. The evolution of true stress vs. time was measured. The relaxation tests were effectuated at six different true strains (from 0.06 to 0.40). The true stress–strain curves obtained at different strains are plotted in Figure 10. As a conclusion, the true stress of PC decreases with time. A great descent of true stress appears during the first 20 s. At the end of the test (100 s), the value of true stress is close to zero.

An exponential equation, including three parameters, was used to characterize the viscoelastic behavior of PC at Tg + 20 °C. The parameters (a, b, and c) in Equation (13) were identified with the least squares method based on the experimental results obtained at six different strains:(13)$$\sigma_{r}=a\times exp\left(-b\times t\right)+c,$$
where $\sigma_{r}$ is relaxation stress, a and b are material constants, and c is equilibrium stress.

The obtained parameters with the coefficients of determination (R^2^) are summarized in Table 3. The value of R^2^ is close to 1, indicating a strong coherence of the obtained parameters with the experimental tests.

The strain rate is zero, because the true strain remains constant in the relaxation test. The viscoplastic strain rate ${\dot{\varepsilon }}^{vp}$ is represented by the following equation:(14)$${\dot{\varepsilon }}^{vp}=-{\dot{\varepsilon }}^{el},$$
where ${\dot{\varepsilon }}^{el}$ is elastic strain rate. Hooke’s law is applied for describing the relation of the elastic strain and stress. A power law is provided to explain the evolution of stress vs. plastic strain:(15)$$\varepsilon^{el}=\frac{\sigma }{E},$$
(16)$$\varepsilon^{pl}=A\sigma^{n_{2}},$$
where E is the elastic modulus, A is strength coefficient, and $n_{2}$ is strain hardening exponent.

${\dot{Ε}}^{el}$ and ${\dot{\varepsilon }}^{pl}$ are expressed as follows, with the viscoplastic strain rate ${\dot{\varepsilon }}^{vp}$ [37]:(17)$${\dot{\varepsilon }}^{el}=\frac{\dot{\sigma }}{E}\;,$$
(18)$${\dot{\varepsilon }}^{pl}=n_{2}A\sigma^{n_{2}-1}\dot{\sigma },$$
(19)$${\dot{\varepsilon }}^{vp}={\left[\frac{\sigma -c}{K}\right]}^{1/m_{1}},$$
where K and $m_{1}$ are material constants. c is found in the quasistatic elastoplastic relation:(20)$$\varepsilon =\frac{c}{E}+Ac^{n_{2}},$$
where ε is the total strain:(21)$$\varepsilon =\varepsilon^{el}+\varepsilon^{pl}+\varepsilon^{vp},$$

Five material constants (E, A, $n_{2}$, K, $m_{1}$) are required for the elastic–viscoplastic model. K and $m_{1}$ are identified based on the true stress vs. viscoplastic strain rate curve. A and $n_{2}$ are identified by the equilibrium stress curves at different strains, as shown in Figure 11, according to the equation:(22)$$ln(\varepsilon^{pl})=ln(\varepsilon -\frac{c}{E})=n_{2}ln(g)+ln(A),$$

.

Table 4 shows the parameters identified from experimental tests for the TLVP model (viscous and plastic parameters) and the elastic properties of the two materials.

## 4. Simulation Results of the Mold Filling in HE

Various behavior laws were applied to PMMA and PC to analyze their effects on mold-filling efficiency in micro HE. The influences of friction coefficient and embossed temperature were also involved in this research. Numerical simulation was performed with the identified parameters as shown in Table 4.

### 4.1. Effect of Imposed Displacement

The elastic behavior was considered in this simulation to observe the filling evolution of PC substrate with different imposed displacements (0.05, 0.10, 0.15, and 0.20 mm). Based on Figure 12a–d, the filling evolution of polymer was represented. As a conclusion of simulation, the maximal von Mises stress was obtained at the mold corner, which was independent of imposed displacement. The lowest value of von Mises stress was 0.62 MPa, obtained with the imposed displacement of 0.05 mm. The highest value was 1.60 MPa, obtained with the imposed displacement of 0.2 mm. The mold filling ratio increased with the imposed displacement, but a micro-gap was unfilled in the process, even with a maximum imposed displacement.

### 4.2. Effect of Plasticity of PC

The elastoplastic behavior (Equations (2) and (3)) was considered in this simulation to observe the filling evolution of PC with different imposed displacements (0.05, 0.10, 0.15, and 0.20 mm). Figure 13 shows the evolution of von Mises stress values of PC substrate in HE. The maximal value of von Mises stress was approximately 0.9 MPa. Because of the plasticity of polymer, the concentration of stress increased in size with imposed displacement, confirmed by the maximum stress zone enlarged with imposed displacement. Similar with the observation in Section 4.1, the mold filling ratio increased with imposed displacement.

### 4.3. Effect of TLVP Model

The elastic–viscoplastic constitutive behavior (TLVP model) of PC and PMMA was applied in this numerical simulation. The material parameters were set based on the value obtained in Table 4. A comparison of the cavity filling ratio with the case of elastic behavior was effectuated to analyze the effects of TLVP model on the deformation of polymer in micro HE, as shown in Table 5. The filling ratio increased as a function of imposed displacements for PMMA and PC.

With the same imposed displacement, the polymer flow filled better in the cavity mold when the elastic–viscoplastic behavior was applied. At the imposed displacement of 0.2 mm, the micro-cavity mold was filled completely in the numerical simulation. It is necessary to consider the elastic, plastic, and viscous properties of PC and PMMA substrates to achieving a filling ratio close to 100% of the mold cavity.

### 4.4. Effect of Friction

The objective was to investigate the effects of the friction between the mold and the polymer substrate on the filling ratio of micro cavity. The numerical and experimental research on the measurement of friction coefficient was realized by Nuño et al. [38], in which the value of friction coefficient µ was from 0.17 to 0.45. In this simulation, the value of 0.4 was selected to study its influence on mold filling. The TLVP model was used to describe the viscoplastic behavior of PC at Tg + 20 °C with an imposed displacement 0.2 mm.

The results of the numerical simulation with and without friction are shown in Figure 14. This figure illustrates the displacement of polymer in the horizontal direction. In the mold/polymer contact area, the displacement of polymer was 5.10 × 10^−4^ mm without friction and 2.13 × 10^−4^ mm with friction (green color). Because the friction between the mold and polymer limited the displacement of polymer, the micro cavity was better filled without friction, confirmed by the smaller empty area shown in Figure 14. It can be concluded that mold/polymer friction affected the filling ratio.

The filling ratio with and without friction was calculated in order to investigate quantitatively its effects. As shown in Table 6, the filling ratio was improved without friction for PMMA and PC. But, it should be pointed out that the difference was quite small. The contribution of friction on the filling ratio was less important than other processing parameters, such as embossing temperature and imposed displacement.

### 4.5. Effect of Temperature

The filling ratio of micro cavity with PC substrate was investigated at different temperature above Tg to analyze its effect during HE. The polymer substrate was embossed at Tg + 20 °C and Tg + 30 °C with elastic and elastic–viscoplastic behaviors in the simulation. The material parameters of polymer at different temperatures are summarized in Table 2.

Based on Table 7, the filling ratio of PC substrate at an elevated temperature was higher than that at a lower temperature when the same behavior law was applied. The filling ratio was higher at an elevated temperature with the same imposed displacement. It can be concluded that the embossing temperature improved the filling ratio of micro cavity.

## 5. Experimental Validation

A uniaxial compressive testing equipment was improved to carry out HE to validate the numerical results. The whole system was placed in a heating furnace to ensure the temperature conditions during the process. An aluminum mold with micro structured features was inserted in the cross head, as shown in Figure 15. The polymer sample was placed on the fixed mold plate. The micro cavity was cylindrical and obtained by micro milling process, with a diameter of 0.20 mm and a height of 0.05 mm, which was coherent with the simulation model. The profile of the micro cavity was measured using the 3D optical profilometer, as shown in Figure 15. Some rugosities were observed on the mold surface, due to the milling process and material.

The polymer sample was placed in the center of the plate to ensure the homogeneity of the distribution of applied pressure. It was polished to reduce the friction between the polymer and mold surfaces. HE was piloted by the imposed displacement and temperature in different steps (embossing, packing, cooling). The evolution of imposed displacement and temperature as a function of time are shown in Figure 16. The mold and polymer were heated to the embossing temperature. The mold insert moved to compress the polymer sample at the temperature above Tg. After the maintain and cooling steps, the mold insert was separated from the polymer sample.

The replicated polymer samples were characterized by 3D optical profilometer Alicona. The obtained profile was compared with the mold insert, as shown in Figure 17. The polymer replica was embossed at Tg + 20 °C, with an imposed displacement of 0.2 mm. The micro feature was successfully replicated on the polymer. The experimental investigation confirmed the numerical results.

## 6. Conclusions and Perspectives

This research concerned the investigation of the viscoplastic behaviors of PC and PMMA polymers and their influence on the filling efficiency of micro HE. The TLVP model was proposed to describe the deformation of polymer above its Tg. The material parameters were identified by inverse method, based on the thermomechanical compression tests. The numerical model was applied in FEM simulation with different temperatures and friction conditions.

The simulation results showed that the filling efficiency increased with the imposed displacement. The proposed TLVP model was confirmed to be more accurate than the elastic model. The micro cavity was completely filled with optimized processing parameters for PMMA (99.99%). The effect of the friction parameter was investigated by considering the viscoplastic properties of PC and PMMA at Tg + 20 °C. The polymer flowed better in the micro cavity without friction. The filling ratio of the microchannel improved with higher embossing temperature. The numerical approach was validated by experimental observation of micro HE, associated with the 3D optical profilometer.

The manufacturing process of an embossing mold could be enhanced by micro electrical discharge machining to guarantee the dimensional accuracy. The behavior model may be improved by taking into account of the effects of molecular weight, polymer chain, and other microstructural features. The optimization of the HE processing parameters will be effectuated to improve the replication efficiency.

## Figures and Tables

**Figure 1 polymers-16-01417-f001:**
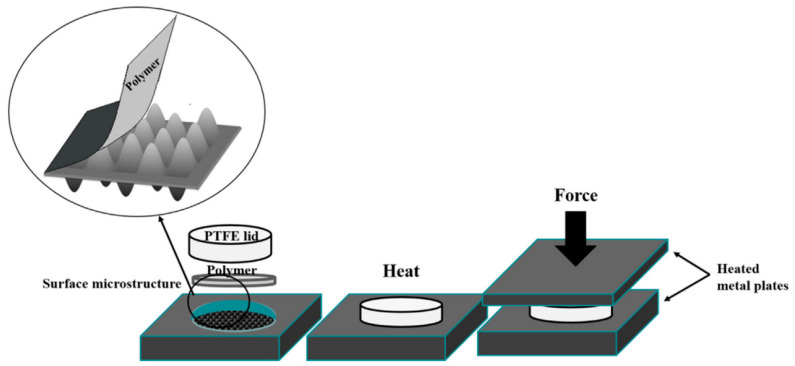
Schematic diagram of HE to replicate micro/nano patterns on polymeric film.

**Figure 2 polymers-16-01417-f002:**
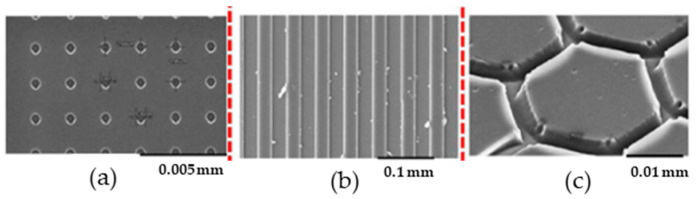
SEM images of (**a**) circular, (**b**) threadlike, and (**c**) hexagonal replicated microstructures obtained via HE.

**Figure 3 polymers-16-01417-f003:**
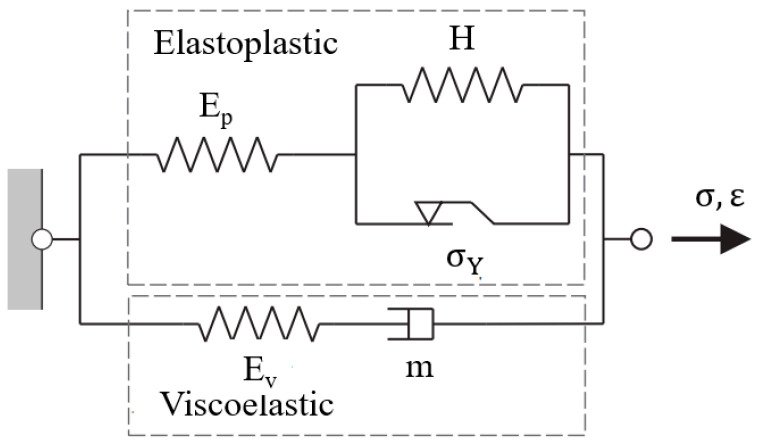
Representation of TLVP model.

**Figure 4 polymers-16-01417-f004:**
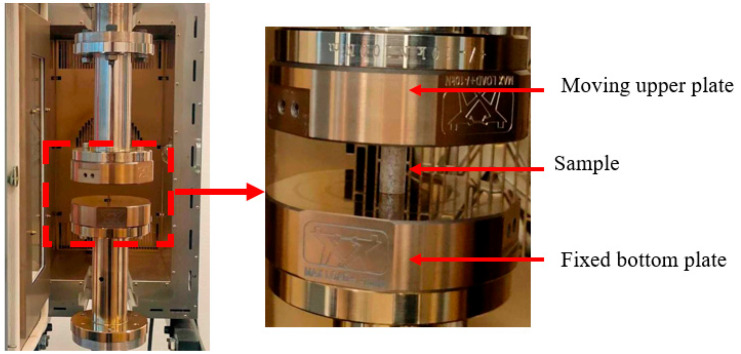
Description of uniaxial compression test with cylindrical polymer specimen (the length of the specimen is 18 mm and its diameter is 10 mm).

**Figure 5 polymers-16-01417-f005:**
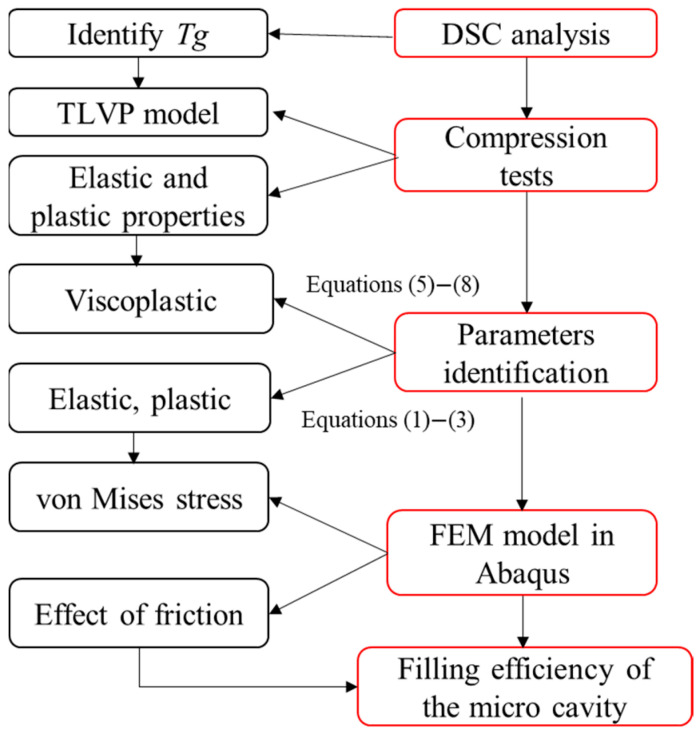
Flowchart to analyze mold filling in HE at micro scale.

**Figure 6 polymers-16-01417-f006:**
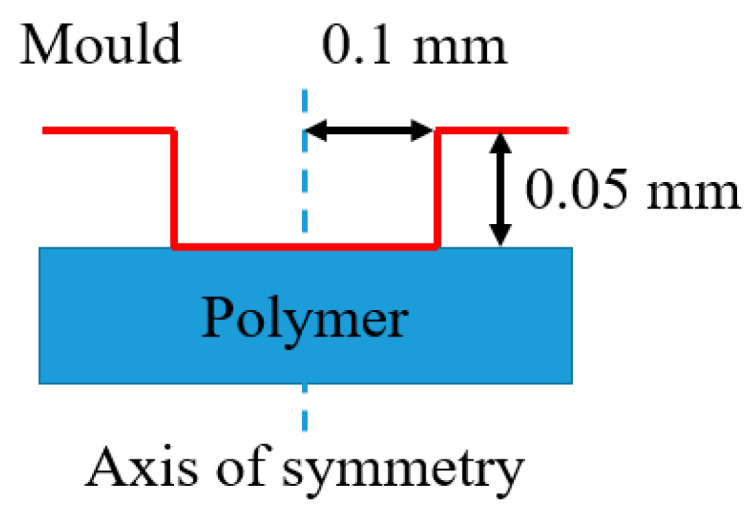
Description of 2D simulation model in HE.

**Figure 7 polymers-16-01417-f007:**
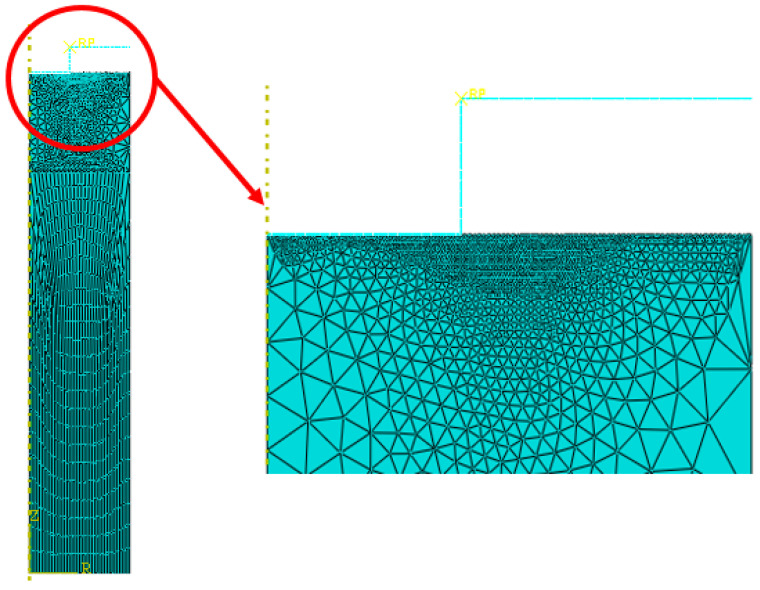
Finite element meshes of the contact zone between tool and polymer used in the simulation model.

**Figure 8 polymers-16-01417-f008:**
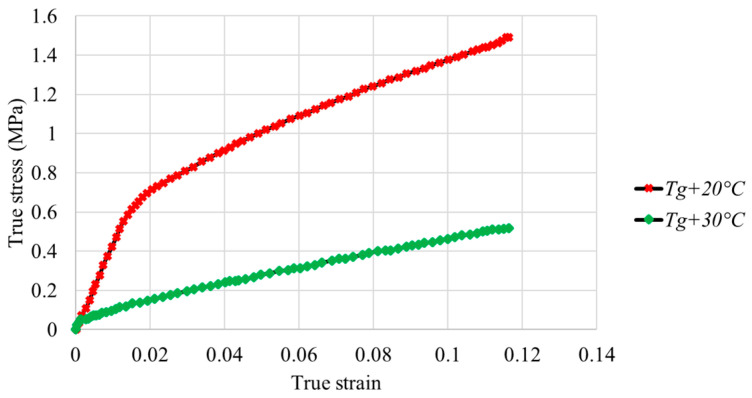
Evolution of true stress as a function of strain in compression tests for PMMA at different temperatures.

**Figure 9 polymers-16-01417-f009:**
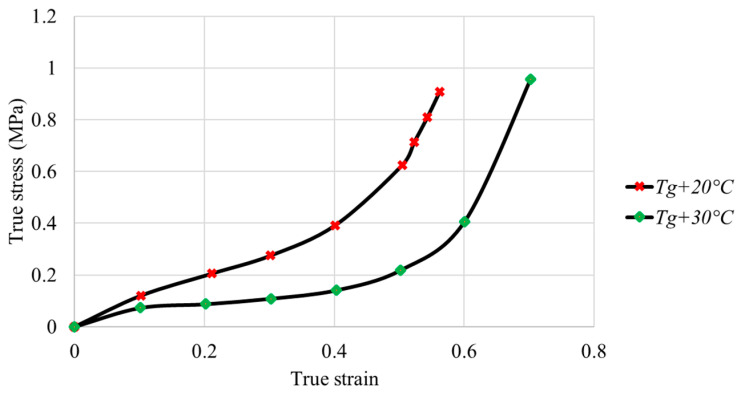
Evolution of true stress as a function of strain in compression tests for PC at different temperatures.

**Figure 10 polymers-16-01417-f010:**
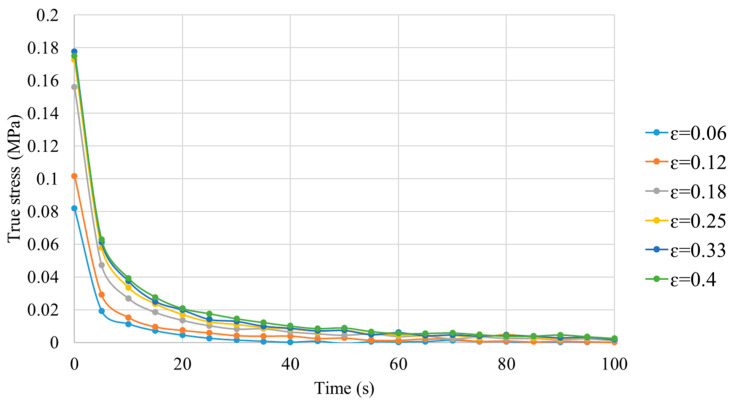
Evolution of true stress vs. time for PC under different strains at Tg + 20 °C.

**Figure 11 polymers-16-01417-f011:**
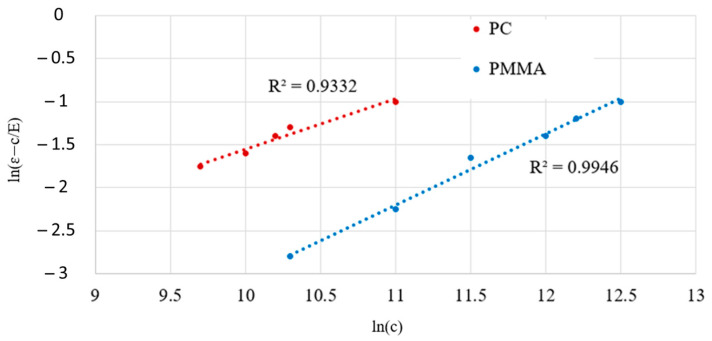
Plastic stain vs. equilibrium stress curves at six strains to identify A and $n_{2}$.

**Figure 12 polymers-16-01417-f012:**
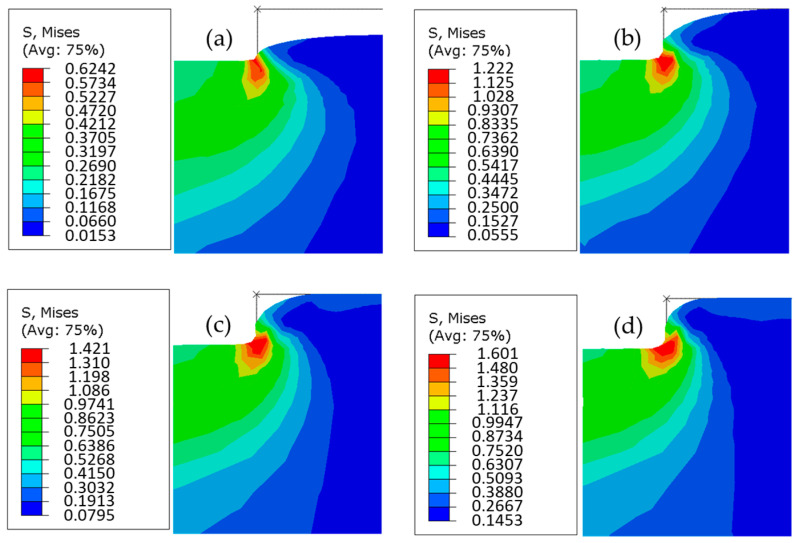
Evolution of von Mises stress in PC at Tg + 20 °C with different imposed displacements: (**a**) 0.05 mm, (**b**) 0.1 mm, (**c**) 0.15 mm, and (**d**) 0.2 mm.

**Figure 13 polymers-16-01417-f013:**
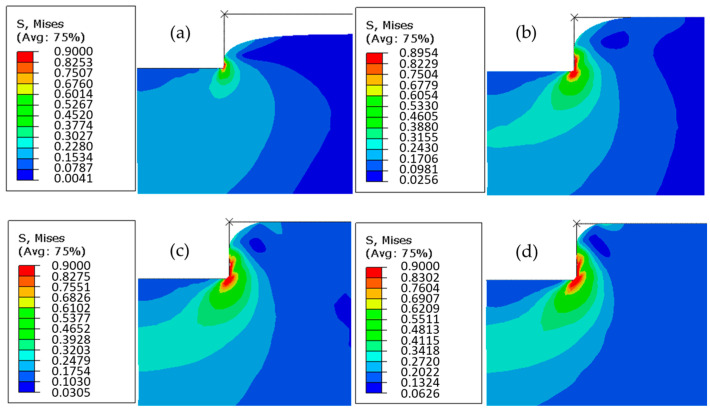
Evolution of von Mises stress in PC using elastoplastic model at Tg + 20 °C with different imposed displacements: (**a**) 0.05 mm, (**b**) 0.1 mm, (**c**) 0.15 mm, and (**d**) 0.2 mm.

**Figure 14 polymers-16-01417-f014:**
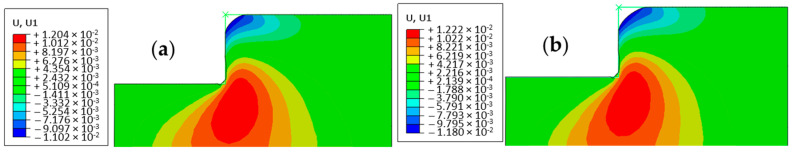
Displacement results for the PC substrate specimen with elastic–viscoplastic behavior at Tg + 20 °C, (**a**) without friction and (**b**) with friction (µ = 0.4).

**Figure 15 polymers-16-01417-f015:**
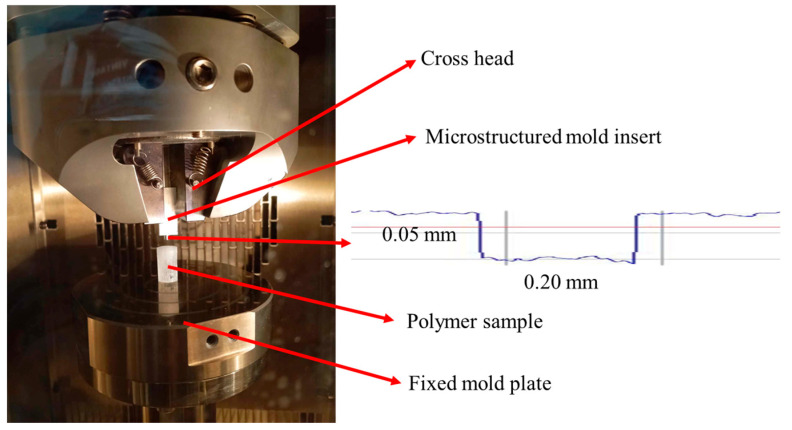
Description of HE equipment and profile of the micro mold manufacturing insert cavity.

**Figure 16 polymers-16-01417-f016:**
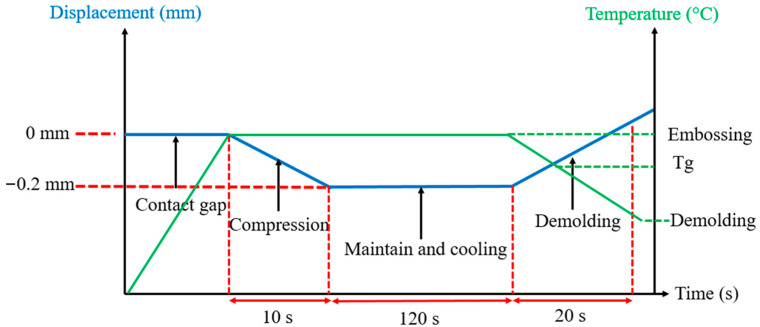
Evolution of imposed displacement and temperature vs. time during HE at micro scale.

**Figure 17 polymers-16-01417-f017:**
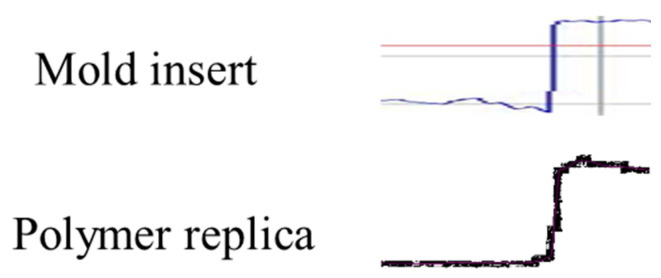
Comparison of profiles between the mold insert and polymer replica obtained at Tg + 20 °C, with an imposed displacement of 0.2 mm.

**Table 1 polymers-16-01417-t001:** Tg of PMMA and PC obtained in DSC tests.

Polymer	Transition Interval (°C)	Tg (°C)
PMMA	112–117	114 ± 1
PC	137–144	140 ± 1

**Table 2 polymers-16-01417-t002:** Elastic modulus, plastic strain, and stress of PMMA and PC at two temperatures.

Polymer	Tg + 20 °C			Tg + 30 °C		
	E (MPa) ±0.005	$\sigma_{p}$ (MPa) ±0.005	$\varepsilon^{pl}$ ±0.005	E (MPa) ±0.005	$\sigma_{p}$ (MPa) ±0.005	$\varepsilon^{pl}$ ±0.005
PMMA	3.61	0.50	0	2.81	0.05	0
		0.80	0.01		0.20	0.03
		1.10	0.04		0.40	0.07
		1.40	0.08		0.50	0.10
PC	2.17	0.12	0	0.97	0.09	0
		0.20	0.05		0.10	0.10
		0.27	0.10		0.12	0.15
		0.60	0.13		0.18	0.20

**Table 3 polymers-16-01417-t003:** Fitting parameters for PC in stress relaxation tests.

Strain	a	b	c (MPa)	R^2^
0.06	0.081	0.1802	0.0010	0.9772
0.12	0.101	0.2201	0.0020	0.9830
0.18	0.153	0.1901	0.0038	0.9797
0.25	0.169	0.2301	0.0040	0.9903
0.33	0.174	0.2001	0.0050	0.9899
0.40	0.170	0.1801	0.0060	0.9944

**Table 4 polymers-16-01417-t004:** Identification of TLVP model parameters of the studied polymers at Tg + 20 °C.

Identification		PMMA	PC
Parameter	Equation		
E_p_ (Pa)	(7)	3.61 × 10^6^	2.17 × 10^6^
E (Pa)	(7)	3.338 × 10^7^	3.173 × 10^7^
f	(8)	0.89	0.93
A (Pa)	(16), (18)	6.63 × 10^−6^	1.71 × 10^−4^
n_2_		0.88	0.70
m_1_	(19)	0.84	0.93
K (Pa)		3.338 × 10^7^	3.173 × 10^7^
v	-	0.4	0.37

**Table 5 polymers-16-01417-t005:** A comparison of filling ratio of PMMA and PC with different imposed displacements from 0.05 mm to 0.20 mm and constitutive behavior laws (elastic and elastic–viscoplastic).

Imposed Displacement (mm)	Elastic		Elastic–Viscoplastic	
	PC	PMMA	PC	PMMA
0.05	43.22%	43.00%	52.10%	98.40%
0.10	76.15%	88.00%	90.50%	98.90%
0.15	89.10%	91.50%	96.60%	99.16%
0.20	97.30%	95.34%	98.00%	99.99%

**Table 6 polymers-16-01417-t006:** Friction effect on the cavity filling ratio.

Polymer	With Friction	Without Friction
PMMA	99.99%	100%
PC	98.00%	98.45%

**Table 7 polymers-16-01417-t007:** Cavity filling ratio vs. different imposed displacements from 0.05 mm to 0.20 mm for PC with elastic and elastic–viscoplastic behavior laws.

Imposed Displacement (mm)	Elastic		Elastic–Viscoplastic	
	Tg + 20 °C	Tg + 30 °C	Tg + 20 °C	Tg + 30 °C
0.05	43.22%	44.12%	52.10%	53.00%
0.10	76.15%	77.15%	90.50%	91.20%
0.15	89.10%	89.96%	96.60%	97.90%
0.20	97.30%	97.83%	98.00%	98.93%

## Data Availability

The raw data supporting the conclusions of this article will be made available by the authors on request.

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
