# Peer review of "Numerical Simulation of Mold Filling of Polymeric Materials with Friction Effect during Hot Embossing Process at Micro Scale"

_polymers, 2024, doi:10.3390/polym16101417_

Round 1

Reviewer 1 Report

Comments and Suggestions for Authors Products made from thermoplastics surround us everywhere, including those produced by the hot embossing method. The presented literature analysis confirms the relevance and importance of the work. On the other hand, citing numerous examples of work and their results, the authors do not clearly formulate their task and goal. I suggest that the authors remove the last lines of the introduction (117-121) in whole and in part. The reader will already understand where in the manuscript the conclusion of the work is... In Figure 1, the inscriptions (polymer abbreviations), etc. are not visible. Lines 128-132 repeat the meaning of the previous paragraph and may be simplified or deleted.   L. 140. “obtained successfully without defects” - the photograph shows obvious defects on the hexagonal edges. Figure 4. I recommend that authors do not use Tg temperature information frequently. At least in this caption this is not necessary. In the text, the authors use mm, and in some figures microns. I recommend fixing it.   The conclusions of the manuscript are consistent with the data presented above. How do the authors take into account the molecular weight of polymers?

Author Response

Products made from thermoplastics surround us everywhere, including those produced by the hot embossing method. The presented literature analysis confirms the relevance and importance of the work.

Q1- On the other hand, citing numerous examples of work and their results, the authors do not clearly formulate their task and goal. I suggest that the authors remove the last lines of the introduction in whole and in part. The reader will already understand where in the manuscript the conclusion of the work is...

R1- Thank you for your comment. The main objective of this research is to propose a suitable viscoplastic behavior law of polymer and improve the simulation efficiency of the HE process. This sentence has been added in the introduction to indicate clearly the goal of the work.

The paragraph (117-121) has been removed.

Q2- In Figure 1, the inscriptions (polymer abbreviations), etc. are not visible.

R2- Thank you for your comment, Figure 1 has been modified to improve its quality.

Q3- Lines 128-132 repeat the meaning of the previous paragraph and may be simplified or deleted.

R3- These lines have been deleted.

Q4- L. 140. “obtained successfully without defects” - the photograph shows obvious defects on the hexagonal edges.

R4- Thank you for your comment. This sentence has been rewritten in the manuscript.

Q5- Figure 4. I recommend that authors do not use Tg temperature information frequently. At least in this caption this is not necessary.

R5- Thank you for your suggestion. “above it Tg” has been deleted in the caption.

Q6- In the text, the authors use mm, and in some figures microns. I recommend fixing it.

R6- Thanks for your comment. The units in figure 2 and 15 have been changed. Only “mm” is used in the manuscript.

Q7- The conclusions of the manuscript are consistent with the data presented above. How do the authors take into account the molecular weight of polymers?

R7- Thank you for your question. The molecular weight of polymers is not considered in this work. But it's a very interesting subject to be investigated in the future. The behavior model may be improved by taking into account of the effects of molecular weight, polymer chain and other microstructural features. This has been added in the last paragraph of conclusion.

Reviewer 2 Report

Comments and Suggestions for Authors

This research focused on the efficiency of microcomponent replication through hot embossing. The presence of unfilled areas often results from inadequate material behaviour models. The study identified a two-layer viscoplastic model for polymers and optimized processing parameters. Mechanical tests were conducted above the glass transition temperature for two polymers (PC, PMMA). Simulation results validated the proposed numerical approach, demonstrating improved filling efficiency with optimized parameters. The effect of friction on filling was also investigated. 

 The authors present interesting article of an engineering nature.
 Enriching the article with a more in-depth description of the phenomena associated with hot embossing is recommended. 
It 
is also recommended to:
- expand the introduction – especially the first paragraph about HE,
- complete the table captions 5 and 7 (more detailed),
- increase the size/quality of figures (1, 4, 7, 15),
- expand and detail the Conclusions and perspectives chapter.

Author Response

This research focused on the efficiency of microcomponent replication through hot embossing. The presence of unfilled areas often results from inadequate material behaviour models. The study identified a two-layer viscoplastic model for polymers and optimized processing parameters. Mechanical tests were conducted above the glass transition temperature for two polymers (PC, PMMA). Simulation results validated the proposed numerical approach, demonstrating improved filling efficiency with optimized parameters. The effect of friction on filling was also investigated.

The authors present interesting article of an engineering nature.

Q1- Enriching the article with a more in-depth description of the phenomena associated with hot embossing is recommended.

R1- Thank you for your comment. The description of HE process is improved in Section 2.1.

Q2- It is also recommended to:

expand the introduction – especially the first paragraph about HE,

R2- Thank you for your comment. The description of HE process is expanded in the first paragraph of Section 1.

Q3- complete the table captions 5 and 7 (more detailed),

R3- Thank you for your comment. The captions of Tables 5 and 7 have been completed.

Q4- increase the size/quality of figures (1, 4, 7, 15),

R4- Thank you for your comment. The size/quality of these figures have been increased.

Q5- expand and detail the Conclusions and perspectives chapter.

R5- Thank you for your comment. Section 6 has been expanded and more details concerning the conclusion and perspective have been added.

Reviewer 3 Report

Comments and Suggestions for Authors

The current work is of added value. Below some minor comments:

L 24 better microstructural control

HE introduction; add a general scheme for a general reader please. Later on one has Figure 1 but this should be earlier.

General comment: check addition of “a” and “the” in several sentences.

Abaqus software; how is this e.g. different from Moldex or other packages?

Can Figure 8 be more supported by literature data, as this seems a more standard plot. I mean Figure 9 has a different configuration.

The discussion can be enriched on PMMA vs PC and how the molecular parameters affect the values.

How can the parameters in Figure 16 be shifted; so can some sensitivity analysis be added.

Comments on the Quality of English Language

see review

Author Response

The current work is of added value. Below some minor comments:

Q1- L 24 better microstructural control

R1- Thank you for your comment. The sentence has been modified.

Q2- HE introduction; add a general scheme for a general reader please. Later on one has Figure 1 but this should be earlier.

R2- Thank you for your comment. A general introduction of HE process has been added in the first paragraph of Section 1.

Q3-General comment: check addition of “a” and “the” in several sentences.

R3- Thank you for your comment. “a” and “the” have been added in several sentences.

Q4- Abaqus software; how is this e.g. different from Moldex or other packages?

R4- Thank you for your question. Abaqus was selected in this research because the proposed TLVP model had been embedded in the software. This saves time in the realization of numerical simulations.

Q5- Can Figure 8 be more supported by literature data, as this seems a more standard plot. I mean Figure 9 has a different configuration.

R5- Thank you for your comment. The obtained true stress-strain curves of PMMA at Tg+20°C are coherent with the analyze obtained by Federico et al. This reference has been added in the manuscript.

Q6-The discussion can be enriched on PMMA vs PC and how the molecular parameters affect the values.

R6- Thank you for your comment. This is a very interesting subject. The authors plan to investigate it in the future. In fact, the behavior model may be improved by taking into account of the effects of molecular weight, polymer chain and other microstructural features. This has been added in the last paragraph of conclusion.

Q7- How can the parameters in Figure 16 be shifted; so can some sensitivity analysis be added.

R7- Thank you for your comment. The parameters shown in figure 16 represent the optimized parameters used in HE process to get the best microfluidic device. The main objective of these experimental investigations is to validate the numerical simulation results. This is the reason why the sensitivity analysis of experimental parameters is not included in the manuscript.
